# Notes towards a Definition of Adaptive Reuse

**Sally Stone**

Manchester School of Architecture, Manchester M1 7ED, UK; s.stone@mmu.ac.uk

**Abstract:** This essay will discuss the evolution of writings about adaptive reuse. The architectural practice is as old as the buildings themselves, yet it has scarcely been discussed or even recognised until relatively recently. The essay will document the varied influences that informed the early publications (the first from 1976). The lack of easily available material (that is, books and documented buildings) meant that pioneering writers had to draw upon other sources—those beyond established architectural discussions. Therefore, these early authors were not limited by the strictures of an already established subject but were able to collate information from a variety of sources. Thus, adaptive reuse draws upon a collage of different sources, many beyond pure architecture, including installation art, fine art, curation, interior design, and urban design. Inevitably, as the subject moves from the periphery of architectural practice towards the middle ground, the number of publications has increased. This diversity has provided the subject with a greater scope, supporting the acknowledgement of the importance of technology, sustainability, and conservation in addition to ideas of heritage and culture, while also allowing for a much less Western-centric focus.

**Keywords:** adaptive reuse; interior architecture; installation art; remodelling; literature; publications

## 1. Introduction

*Fragments of a vessel which are to be glued together must match one another in the smallest details, although they need not be like one another. In the same way a translation, instead of resembling the meaning of the original, must lovingly and in detail incorporate the original's mode of signification, thus making both the original and the translation recognizable as fragments of a greater language, just as fragments are part of a vessel.*

Walter Benjamin [1] (p. 260)

For such a long-established and deeply entrenched subject, adaptive reuse has a remarkably short history. It is a practice that stretches back to almost the first constructed buildings themselves, for structures have perpetually been altered to accommodate the needs of their different occupants [2] (p. 114), and yet it has continually lacked the written theoretical and historical recognition of new-build architecture. This could be based upon the prejudice inherent in the concept that interior design (as the practice was once called) was regarded as a respectable profession for women, combined with the lack of perceived worth in adaptive reuse within the modernist and late-modernist world. However, the agenda of the 21st century has ensured that adaptive reuse is beginning to be accepted as a professionally relevant and creative method of developing the built environment. This builds towards this century's environmental urge to adapt and transform combined with the need to build human experiences, rather than construct new things. The current mantra "reuse, reduce, recycle" is an indication of this massive shift in attitude. Adaptive reuse is now seen as one of the most significant issues within the architectural profession.

The act of working with the already-built implies compromise, it suggests that the designer, rather than imposing their own vision upon a specific place, must first understand the agenda of the building before presuming to change it. This implies negotiation, agreement, and conciliation. But adaptive reuse is also transgressive; it undermines the

primacy of the original architect, makes secondary his or her agenda, and overlays this with new meaning.

The recognition of the practice of adaptive reuse as an appropriate architectural approach is still so new that an exact title has yet to fully emerge (also, an exact definition). Adaptive reuse does seem to be evolving into the settled term for the subject, however, the practice can also be referred to, among other options, as: interior architecture, remodelling, building reuse, retrofitting, conversion, adaptation, rehabilitation, reworking, refurbishment, or, sometimes, especially in North America, repurposing. These are reverential terms, possibly transgressive or subversive, but they do not exhibit overt authority, so it is interesting that the website 'Building on the Built' refers to the practice as 'interventional work'; these are both assertive words that seem to elevate the approach into a much more proactive, definite, and less deferential activity.

Certainly, there are many terms to describe the process; Graeme Brooker and Sally Stone's ReReadings described it as *interior architecture* [3], while Fred Scott and Philippe Robert both use the single word *adaptation* [4,5]. Even as late as 2022, there was still ambiguity about the title and the definition; Francesca Lanz and John Pendlebury in a 2022 essay, called Adaptive Reuse: A Critical Review, declared that… "*there is no common and shared agreement on what adaptive reuse precisely is and what it entails*" [6]. Fred Scott describes the subject as *alteration*, which he defines as the "mediation between preservation or demolition" [4]. Bie Plevoets and Koenraad Van Cleempoel propose that it is *altering existing buildings for new or continuous use* [7]. Johannes Cramer and Stefan Breitling suggest that it is *architecture within existing built contexts* [8]. James Douglas: *Any work to a building over and above maintenance to change its capacity, function or performance* [9]. The ICOMOS definition is: *Adaptation means the processes of modifying a place for a compatible use while retaining its cultural heritage value. Alteration processes include alteration and addition* [10]. While the Burra Charter says that *Adaptation means changing a place to suit the existing use or a proposed use* [11]. Sally Stone's monograph UnDoing Buildings suggests that adaptive reuse is described as utilizing *strategies that are applied not as a reaction but in anticipation* [12]. Phillipe Robert elaborates to suggest that it is the *story of the successive layers, of the reshaping of monuments, and of the additions that bear testimony to each succeeding age* [5], and Frank Peter Jager simply states *work with existing buildings* [13].

> *The past provides the already written, the marked 'canvas' on which each successive remodelling will find its own place. Thus the past becomes a 'package of sense' of built up meaning to be accepted (maintained), transformed or suppressed (refused).*
>
> Rodolfo Machado [14]

## 2. A Review of Sources

This is an opportunity to dwell upon the evolution of the subject and to discuss the literature that has informed the approach, plus the buildings, architects, artists, and installations that have contributed towards that formation. Given that adaptive reuse has evolved into one of the predominant aspects of architecture in the 21st century, it would be interesting to approximately divide this study by the turn of the millennium.

## 3. Common Link

The common link throughout this discussion is the connection with place. The idea that the authenticity of place, the reality of a tangible situation, and the sensory connection with the actual physical certainty of somewhere substantial and quantifiable can not only create a connection with the past, but can also generate a new future. In an anxious world of continual surveillance, with virtual realities that are not necessarily real and truths that are not completely true, this connection with the actual physical situation of a definite place provides a sense of certainty that is often not readily available elsewhere. Adaptive reuse creates a real connection with place. The relationship is real and tangible, it is authentic and contains certainty.

Adaptive reuse responds to the situation to which it is directly connected. This is both a tangible physical connection with the material reality of the environment, but also with the intangible collection of forces that formed it. Whether these are cultural, climatic or geographical, man-made or natural, they are elements that comprise the situation of the place, they inform its character and the way in which people react to it. The crisis within the contemporary city means that continued horizontal development can no longer be supported, but the built environment needs to build in on itself, to be more dense, more productive, more resilient. Buildings, situations, and neighbourhoods are in a continual state of flux, they are altered, updated, maybe rehabilitated, but rarely do they exist in a state of scarification. This incessant renewal is an opportunity to accommodate the needs and aspirations of those who occupy the place, hopefully before they even realise that they need the change. The connection with place is exemplified by how intricately these elements are linked together, how this connection can create a ripple through the continuity of existence, and how it can build a better future.

## 4. Before the Millennium: A Collection of Texts, Buildings, Interiors, and Installations

There are two factors that tie together this collection of texts, buildings, interiors, and installations. The first is that intrinsic relationship between the built form and the environment that it inhabits. Contextualism (which is a design tool/approach rather than a style) connects all the sources discussed. Adaptive reuse projects enjoy the double dialogue of the context of surrounding area of occupation, plus the conditions of the host building.

The second factor is the attention to detail. The original building is an intense collection of tangible and intangible elements, an assembly of real and virtual parts that gather together to form a coherent image of the structure, and it is to these that the interventions of reuse render respect and respond. To be able to alter a building, the designer needs to develop an intimate understanding of individual parts. The list of areas of understanding is long: materials, methods of construction, structural system, rhythm of spaces, position of the openings, circulation, and many more, all of which contribute to the physical conditions of the existing building. This is combined with more intangible components connected with the culture of those who first constructed the building, those who occupied it, and those who will occupy it. To develop an intense dialogue between old and new, the designer needs a personal relationship with the old and the new on an intimate scale, for these are not abstract buildings on greenfield sites, but they are an exquisite uniting of two allied but not identical individual surfaces that flex and deform, soften and align, to create a union of convergence.

## 5. Significant Publications about Adaptive Reuse

By the beginning of the 21st century, there were books full of case studies, books that discussed the practicalities of the subject, picture books that tickled the surface of the subject, books that just about included the area as a periphery to the focus of the discussion, and essays that touched on it, but just two publications that systematically analysed and discussed the process, approach, or methodology of adaptive reuse: Machado's *Old Buildings as Palimpsest* and Robert's *Adaptations*.

Probably the most relevant is Rodolfo Machado's *Old Buildings as Palimpsest* [14]. The U.S.A. journal *Progressive Architecture* published the four-page essay in 1976; initially it was relatively unknown but over the last half-century it has become a recognised approach and indispensable source. Machado's use of the palimpsest as an analogy for the process of adaptation perfectly describes the pluralism inherent within the approach. The essay also introduces the concept of "form following form". Machado declared that "... *the form/form relationship is the primary consideration within remodelling activity*" [14]. This turned the prevalent mantra "form follows function" on its head. The idea that the influence of the enclosing buildings is so great that it becomes the primary driver for the methodology of reuse was revolutionary and far from the prevailing idea that the relationship between old and new was secondary to the proposed function and the ego of the architect.

The other important publication was *Adaptations: New Uses for Old Buildings* by Philippe Robert [5]. It was initially published in the *French Architecture Thematic Series* by Editions du Moniteur in 1989 and was translated into English by Murray White and published by Princeton University Press in 1991. The book is radical, it broke new ground, it was progressive, and, as the front runner, very important. It is organised in a simple tri-part order: 1. Introduction; 2. Detailed case studies illustrated with photographs, measured drawings, and sketches; 3. A small catalogue of recently constructed landmarks.

The very short introduction is rich and powerful. Conversion, the author declares, can be considered as a "*normal architectural practice*" [5] (p. 6)—a distinctly early proclamation for what is now a ubiquitous approach. Later in the same paragraph, Robert asserts that the renewed awareness of the history of architecture includes the "*history of buildings that have been altered*". This, of course, coincides with the post-modern ideas of the return of history, the importance of the individual, the embracement of pluralism, and the search for eclecticism.

Within the density of the introduction, Robert also discusses the idea of the palimpsest as a metaphor for adaptive reuse. Machado is not listed in the bibliography, but this idea, and that of the form–form relationships, are analysed. In fact, Claude Soucy is listed as the source for this and quoted thus: "*Out of the encounter between old envelope and new requirements and means, a unique object will be born-one which is no mere juxtaposition, but a synthesis from the point of view of both construction and architecture*" [5] (p. 9).

An innovation in the intense opening chapter is the classification of the different approaches to adaptation. Robert lists seven strategic types of approach: building within, building over, building around, building alongside, recycling materials, adapting to a new function, and building in the style of. This inventory is intriguing and, yet, also unwieldy and messy—the taxonomy seems too ambiguous; it lacks focus and appears incomplete. Despite the creative originality in the classifications, it is difficult to place a number of key buildings within any category. For example, a more exact group is needed to house Scarpa's masterpiece Castelvecchio Museum (Verona, 1956–1973) and his ethereal Querini Stampalia Foundation (Venice, 1963), a group that would recognise the scraping away of parts of the building and the addition of a series of new elements. The Tate Modern (Herzog and de Meuron, London, 1999) is difficult to classify, as is the Irish Film Centre (O'Donnell and Tuomey, Dublin, 1992), which is the conversion of nine connected buildings.

There are a couple of other books that are focussed upon adaptive reuse, and, certainly, at a time of scarcity, were significant, but this has faded over time. The 1989 publication *Re/Architecture* by Sherban Cantacuzino [15] is a beautifully illustrated book with over 50 case studies. It contains six chapters and is organised by the function of the original building, so, for example, the first chapter examines public buildings, and the first case study is the conversion of the Helsinki City Hall for multi-use purposes, and the second is the transformation of the Gare d'Orsay in Paris into a national museum. Each chapter is prefaced by a well-informed and accessible introduction. The main introduction to the book discusses the importance of the stock of existing buildings as an opportunity for urban regeneration, but also useful for "*sound economic, social and ecological reasons*" [15] (p. 9). A far-sighted prophesy indeed! Kenneth Powell's *Architecture Reborn: The Conversion and Reconstruction of Old Buildings* [16] takes a similar approach to the structure of the book but uses the transformed function rather than the original use as the subject for each chapter. The book, as would be expected from someone with such a reputation, is very well researched and engagingly written; it is big and dense, with well-produced photographs and supported by architectural drawings. Powell's introduction, as Cantacuzino, begins with an historical survey, but ends with a call to arms. "*The issue is no longer about new verses old*", he declared, "*. . . but about the nature of the vital relationship between the two.*" The introduction concludes with an assertive quote from David Chipperfield: "*We must inhabit an ever-evolving present, motivated by the possibilities of change, restricted by the baggage of memory and experience*" [16] (p. 19).

## 6. Other Publications That Discussed a Contextual Methodology

Post-modern pluralism, new urbanism, and contextualism all played an important role in the rise of a specific adaptive reuse theory. There were a number of highly influential books; these did not discuss adaptive reuse per se, but certainly included it among the searching ideas for a new urbanism. Seminal publications, such as Jane Jacobs' *The Death and Life of Great American Cities* [17], Colin Rowe and Fred Koetter's *Collage City* [18], Aldo Rossi's *Architecture of the City* [19], Robert Venturi's *Complexity and Contradiction in Architecture* [20], Michael Graves' guest editorship of the *Roma Interrotta* project [21] edition of *Architectural Design*, and Thomas Schumacher's short essay *Urban Ideals and Deformations* [22] all promoted the idea of the city as an eclectic mix of old and new that could together create a progressive and harmonious future. All of these ideas dealt with the development and redevelopment of the existing built environment, and, so, were easily extended to the adaptation of existing buildings.

Graves's editorship of *Roma Interrotta* of 1979 documented the project that was invented and developed by Piero Sartogo, which took the breath-taking Nolli Plan as inspiration. Sartogo asked 12 prominent architects to reimagine it, each taking a proportional section of the great drawing as both the starting and the finishing points. So, the drawing was complete at the beginning of the process, and whole, once again, at the end, but, as a palimpsest, it had been rubbed away and redrawn during the course of the project. The Roman Interventions interrupted the drawing; they did not obliterate the grain of the city, the organisation of the streets and squares, the position of the buildings, and the arrangement of the interiors, instead, the architects worked with these attributes, producing what was then a radical fusion of old and new.

Colin Rowe was one of the guest contributors to the reimagined Nolli Plan who, together with Fred Koetter, had published *Collage City* just a year earlier. This narrative discussed the crisis within the modern city, the problems connected with the obliteration of history and the need for a more contextual approach to architecture. The book, which starts as a methodical undoing of the prevailing attitudes towards architecture and urban design, continues as a call to arms for a new approach, and ends as a handbook of inspirational approaches to guide the way forward.

Rowe and Koetter discuss such romantic suggestions as the apotheosis of the collision, the search for bricolage, and the reconquest of time. A significant discussion is the comparison between le Corbusier's monumental *Unité d'Habitation* and Giorgio Vasari's *Uffizi Palace in Florence* [18] (p. 69). One is the inverse of the other, so, while the Unité is a solid monolith, so the Uffizi is a void. Thus, the area of land surrounding the Unité is deformed to accommodate the regular building, and, conversely, the building surrounding the void, or Vasari's Corridor (as it is known), is deformed to accommodate the space, so undermining the modernist ideas of the primacy of form and opening up the possibilities of building in and around existing structures.

The idea of *Contextualism: Urban Ideals and Deformations* was further explored by one of Rowe's students, Thomas Schumacher [22]. He treads very much the same path as his tutor, but in nine intense pages that call for some sort of middle ground between an artificial incarnation of the past and the brutalising and dominating system of modernism [22] (p. 297). An ideal form can exist as a fragment "*collaged*" into an empirical environment [22] (p. 301). Contextualism, he asserted, is a design tool that could be abstracted to any given situation. Kate Nesbitt who collected the essay her edited collection: Thoerizing a New Agenda for Architecture [23] (p. 294), recounts that Schumacher's recollection was that Contextualism is a conflation of Context and Texture. The term he suggested, was first used by Steven Hurtt and Stuart Cohen.

Venturi and Scott Brown's *Complexity and Contradiction in Architecture* [20] uses historical precedents to propose a methodology for moving forward; an attitude that suggests that everything is valid, that there is a need to move away from the tabula rasa approach, and, even more so, away from the primacy of the monumental volume. The opening section, entitled Nonstraightforward Architecture: A Gentle Manifesto, called for elements that

are "...*hybrid rather than pure, distorted rather than straightforward, ambiguous rather than articulated...*" [20] (p. 16). (Robert Venturi is listed with the authorship of the book, but Denise Scott Brown's contribution is now so recongnised that she is normally credited as co-author.) Especially relevant to this discussion is chapter 9, on the importance of the interior, and the understanding that the exterior and the interior could have different personalities; that the interior is much greater that the mere consequence of the containing exterior walls. They railed against the modernist orthodoxy of the continuity between the inside and the outside, that one should slip easily into the other to create a "*oneness*" [20] (p. 301). (This attitude later caused the great interior theorist Fred Scott to suggest that "*the interior had escaped from the building*" [12] (p. 15)).

Venturi and Scott Brown supported the idea that the exterior and the interior of a building could be different, and this separation, he argued, emphasised the identity of both. They reasoned that contradiction may be further emphasised through the use of detached linings, which can leave spaces between the structure and the interior, thus providing opportunity for interpretation. They were not explicitly discussing adaptive reuse but more providing the springboard for further consideration. This exploration of difference was an incentive for remodelling. Importantly, this is greater than a book about urbanism, it is about the comfort of enclosing space rather than the significance of epic building, a pursuit of modesty, about the understanding of how a collection of intricate details can create a greater whole, of how the environment of the already-built could provide the impetus for future development, and how all of these could appear to have always been there but are so obviously of the now.

Another significant publication that opened as an attack on the principles and aims that have shaped modern, orthodox city planning and building, then evolves into a manifesto for an exuberant and diverse city, is Jane Jacobs' *The Death and Life of Great American Cities* [17]. Her far-sighted call for the new to mingle with the old, for building to address the street, for places to serve more than just one primary function, and for density of population [17] (pp. 150–151), is now acknowledged as an astute recognition of how to address the 21st century concern with sustainable population growth in cities. Some 60 years after publication, densification conducted through the adaptive reuse of the stock of existing buildings is the established approach to development.

These texts were important to the development of a methodology for adaptive reuse, they regarded the built environment as an evolving situation of discourse, and the ideas developed and discussed were as relevant to individual buildings as they were to larger urban environments.

## 7. Art, Architecture, and Design

As important as the publications was the work of specific architects and designers who pursued a contextual approach in their work combined with a love of heritage and history; they also searched for narratives and fables, and wrapped this in a post-modern sensibility. Visual people habitually spend longer looking at the pictures than reading the words, so buildings displayed in such journals as *Blueprint* or *Journal of Interiors*, combined with visits to the places, were often more important than the texts that discussed them. Architects and designers included Carlo Scarpa, Group 91, O'Donnell and Tuomey, Hans Hollein, John Outram, Nigel Coates, James Stirling, Vittorio Gregotti, Aldo Rossi, Robert Venturi and Denise Scott-Brown, Ron Herron, Aldo Van Eyck, Rafael Moneo, Coop Himmelblau, Ken Belly, David Chipperfield, Memphis...

Carlo Scarpa is regarded as the master of adaptive reuse, yet his work was often acknowledged as lacking architectural intent. The practice of adaptive reuse has long been seen as having limited worth and beneath the interest of many architects. Even as late as 1993, Richard Murphy, in his highly detailed and intense discussion of the Fondazione Querini Stampalia Foundation (Venice, 1963), questioned the veracity of the design and asked whether Scarpa's work was "*merely interior design*" [23] (p. 3).

And yet, young architects were beginning to establish a reputation with such projects. David Chipperfield's early shop interiors (for example, the Issey Miyake boutique, Sloane Street, London, 1985) were formative little projects that combined careful craftmanship with an exploration of complex spatial relationships. Hans Hollein created a series of daring, yet refined, individual shops in Vienna (for example, the Retti Candle Shop, 1966, or the Schullin Jewelery Store, 1974). These long, narrow stores that appear slotted into the available spaces are exquisitely executed interiors—as would be expected in the birthplace of the Secessionist movement. The great documenter of post-modernism, Charles Jenks, wrote a rapturous review of Hollein's early work: "*So much design talent and mystery expended on such small shops would convince an outsider that he had at last stumbled on the true faith of this civilisation*" [24] (p. 32).

The radical post-modern architect John Outram is recognised for the ground-breaking Pumping House in the Isle of Dogs (London, 1986). Outram's buildings are borne from ancient myths and modern parables and invoked the inherent romance of Claude Lorraine's landscapes; the pumping station in the Isle of Dogs conceptually contained columns that penetrated hundreds of feet through the mud and silt to connect with the bedrock, while the roof of the Kensal Road housing swoops gracefully from above to gently land upon the building. He also completed the transformation of an ordinary two-storey 1960s concrete-framed office block into an articulate yellow brick-clad building which was seemingly supported by great bulbous columns with flaming capitals. But it was cleverer than a mere cosmetic revamp. All of the services were diverted into ducts hidden within the fat columns, and those that did not contain such facilities performed other useful services—the coffee machine, the filing cabinets, the fire extinguishers.

The Temple Bar Framework Plan in Dublin by Group 91 [25] (Dublin, 1996) was equally influential. The substantial area next to the River Liffey had been earmarked for a huge bus station; in fact, in 1977 Skidmore, Owings & Merrill Architects produced a scheme for a great spiraling monolith to completely fill the site. When, years later, this proposal was abandoned, the city council held an architectural competition for the complete neighbourhood. The winning project proposed to regenerate the area through the construction of a series of cultural buildings, which would tuck into the urban grain of the area, thus allowing the natural rhythm of the place to be retained. The scheme proposed a mixture of new buildings and adaptations, the most notable being the Irish Film Centre. O'Donnell and Tuomey were individually responsible for this amalgamation of nine different existing buildings set deep within the city block.

Another seminal adaptation and, perhaps, the last to mention here is the Haçienda (Manchester, 1982)—once described as the "*most famous night club in the world*". Ben Kelly's joyful, post-industrial, post-modern approach to adaptive reuse has proved to be absolutely revolutionary, and his paradigm-changing design for the interior of the nightclub has become part of a powerful cultural legacy rooted in both the city's and the era's industrial aesthetic; it has proved to be internationally influential.

It is also important to discuss the influence of installation artists to the development of adaptive reuse. Artists can experiment with existing buildings and spaces without the pressure of the needs of the end users and the exacting regulations connected with construction, therefore, they are often in the position to push ideas further and more quickly than the architect or designer is able.

The Gordon Matta-Clark retrospective at the Serpentine Gallery in London, 1993, [26] was a timely and powerful exploration of the impact that considered dissection can have upon existing buildings. Matta-Clark cut holes in buildings, whether to create connections that did not previously exist, to reveal unfound associations, and in one piece, Splitting, he actually cut a timber house in half, to expose the flimsy insubstantial nature of the structure and, maybe, also of inhabitation itself. The exhibition caught the mood of many architects and designers of the time who were beginning to question the dominance of new buildings when perfectly good strong and useful ones still existed, of the removal of built heritage, and the prevailing lack of legacy that resulted. The other important aspect of Matta-Clark's

work was the conceptual idea of the subtraction of material. This was tantamount to an anti-heroic architectural move; it was exactly the opposite of the progressive and productive monument to the exaltation of the architect.

There was another installation of equally massive impact the same year as the exhibition; House by Rachel Whiteread (London 1993). This installation uncovered the actual space within the interior of a single house in a soon-to-be-demolished terrace in London. Whiteread used the structure as a mould to create a three-dimensional representation of the interior of the rooms by spraying the inside of the exterior walls with concrete then removing the walls, thus leaving the insides exposed. This included the reverse of the mouldings around the doors and windows, the reverse of the windows, and, significantly, the patina of time and use on the walls themselves.

The artists, although a generation apart, were equally radical. They questioned the substance from which buildings were constructed, and, by extension, the basis of the society that constructed those buildings. The exposure of the insides of the House was deemed to defile the people who had once lived there, while Splitting was seen as a comment about the insubstantial lives of those who occupied it. Despite their shock appeal (and, by the end of the 20th century, it was getting very difficult to shock people) their work was beautiful, poised, and knowing—about architecture, structure, balance, and life.

There are other artists who were also important, these include Cornelia Parker, whose installation Cold Dark Matter: An Exploded View is the exposed violence of a detonated shed. Again it uses an existing structure and the beauty of the resultant installation creates impact and questions the strength and permanence of the built structures around us. Robert Irwin (*There is No There There until You See There There*), James Turrell who created exquisitely clever installations with pure light, Richard Wilson whose installation of a huge treacherous tank of thick dark reflective oil in the Sachi Gallery gave the space an ambiguous shape and size, and Alison Turnbull, whose manipulated images were generated by seemingly randomly discovered architectural drawings that were then expanded, revised, changed and subjected to alterations that, like the palimpsest, retained the essence of the original, but created a completely new proposal. These artists explored existing spaces and forms, then attempted to heighten the impact of these given places through considered interventions.

## 8. After the Millennium: An Examination of the Canon—Books about Adaptive Reuse

These conditions generated a collection of publications that have begun to create a canon of thought about adaptive reuse. Given the relative youth of the subject, the books are spare and focussed, but it is interesting to observe that, as the 21st century progresses, how the breath is being discovered. Over the last 20 years, the number of books specifically about adaptive reuse has proliferated. The majority of these can be easily divided into two categories: those that make extensive use of case studies to illustrate themes or processes (Frank Peter Jäger, Christian Schittich, Graeme Brooker and Sally Stone, David Littlefield and Saskia Lewis, Johannes Cramer, and Stefan Breitling), and those that carefully build the argument through a series of illustrated discussions or chapters (Fred Scott, Lilian Wong, Sally Stone, Bie Plevoets, and Koenraad Van Cleempoel).

## 9. Case Study Books

It is inevitable that the case study books should use a similar organisational approach to those published before the turn of the millennium, but the focuses of the studies differ; from quite technical explorations, through poetic interpretation, to books that shout about the urgency of the situation. The system of classification, rather like a translation, is always partial and emphasises the interests and obsessions of the author(s). This subjective process of taxonomy is determined by the culture and experiences of the individual(s) who make the selection, thus, there are both different selections of buildings and different interpretations of the chosen buildings. Keith Jenkins explains that the basis of this emphatic system of interpretation are the morals imposed by contemporary society, and that "...*given that interpretations of the past are constructed in the present, the possibility of the historian being able to slough off his present*

*to reach somebody else's past on their own terms looks remote*" [27] (p. 40). To extend this further, the preoccupations of the authors guide the taxonomic process. This is doubly complicated, as the process of bringing a building from a past existence into the present can be seen as a work of translation (Scott, Stone, Van Cleempoel), as the inspirational equivalent to transcribing from one language to another.

It is within the discussion of the introductions that the differences are revealed. Jäger, whose criteria for case study selection is architectural quality, describes the process as "*A Gift from the Past*" [28] (p. 11), Schittich, whose selection is deliberately optimistic, describes it as "*Creative Conversions*" [29] (p. 9), Cramer and Breitling pursue clarity—in both the intellectual process and construction techniques [8] (p. 9), Stone and Brooker pursue an architectural approach [3], while Littlefield and Lewis place adaptive reuse among a great artistic tradition of decay and rebirth [30] (p. 15).

Brooker and Stone's *ReReadings: The Principles of Interior Architecture and the Reuse of Existing Buildings Volume 1* (2004) was probably at the vanguard, but there are significant books not far behind. The book, which builds upon a synthesis of the pre-21st century texts and precedents, was at the forefront of an oncoming movement that placed much greater emphasis upon the already-built, that valued history and heritage, that used a post-modern sensibility to create a new future that learnt from the past but, equally, considered the need and aspirations inherent in the future. But, unlike much of the previous literature, *ReReadings* presented a methodology for the future of the already-built. *ReReadings* assembled the collection of impulses and arranged them in a comprehensible order that rendered the process accessible to all involved. It set this out in easy stages, and, so, a process that had previously been seen as slightly impenetrable, complicated, and difficult to read was rationalised.

When the first volume was published, it broke new ground, it proposed ideas that, although part of a continuity, were quite radical. The urgency of the book combined with the lack of easily available information means that it has a very Western focus. There were few case study projects beyond Europe and the U.S.A. The reflection made possible by the decade and a half between the publication of the two volumes in the series allowed for a more relaxed and inclusive approach to the selection of case studies in *ReReadings: The Principles of Interior Architecture and the Reuse of Existing Buildings Volume 2* (2018). Diversity is demonstrated through the selection of the projects; this was an opportunity to expand the geographical areas discussed. The focus is still predominantly European, but projects in Malaysia, China, Japan, Taiwan, Brazil, and Australia are also presented. Discussions of sustainability, digital and other methods of occupation, plus a much less Western-centric selection of case studies, pushes the argument beyond the normal bounds of architecture and interiors, and embraces many of the cross-disciplinary and diverse aspects of the subject.

*Building in the Existing Fabric: Refurbishment, Extensions, New Design* (2003), edited by Christian Schittich, expresses very similar sentiment, that a turning point in the attitude towards existing buildings has been reached and conversions are, the author declares, the "*New Normal*" [29] (p. 9). He suggests that conversion and renovation are no longer seen as a "*necessary evil*", that things have changed, and the process has become one of the "*most creative and fascinating tasks in architecture*". The introduction acutely states that "*For a long time, Carlo Scarpa's refurbishment of the medieval Castelvecchio in Verona (1956–1964) was considered the benchmark for all creative conversions*" [29] (p. 9), and also suggests that none of its vitality has been lost. There are 24 good examples of adaptive reuse described, and each is accompanied by good photographs and detailed drawings. In *Architectural Voices: Listening to Old Buildings* (2007), David Littlefield and Saskia Lewis, again, catalogue a well-researched and insightfully described collection of case studies which are prefaced by a nostalgic introduction that almost wistfully looks for traces of romance within the history and patina of the existing structure. The bibliography reinforces the lack of available literature in these first few years of discovery, with just one book about adaptive reuse [30]. Frank Peter Jäger's *Old & New—Design Manual for Revitalizing Existing Buildings* [28] utilises

three groupings: addition, which discussed new elements within or around the existing; transformation, which represents a change of appearance; and conversion, which denotes a change of use. The case studies are well illustrated, the discussions certainly have technical depth, and there is an emphasis on projects that contain aspects of the socio-political, but the classifications do seem somewhat arbitrary. Johannes Cramer and Stefan Breitling in *Architecture in Existing Fabric: Planning, Design Building* (2007) [8] use case studies grouped into chapters each with in-depth discussion. The order of these is ingeniously dictated by the design process itself, so the chapters are, from the beginning, Planning Process, then Preparatory Investigations, followed by Design Strategies, Detail Planning, Building Works, and concluding with Sustainability, which, if the truth be known, and however important the subject is, does feel a little bit like an add-on at the end. The series, *Basics Interior Architecture: An Approach* [31–33], again by Brooker and Stone, also fits into this chapter-driven illustrated case study category. The methodology for these is informed by the analysis of the existing situation combined with a developed approach to the changes proposed by the architect. The process of reading an existing building can be divided into three basic categories: *Form and Structure* (2007, reprinted 2016), *Context and Environment* (2008), and *Elements and Objects* (2010). Suzie Attiwill describes the organisation of Brooker and Stone's *ReReadings* (and this could also apply to all of the books in this section) as rather "*like a curated exhibition, various examples are selected to illustrate each category*" [34].

## 10. Books That Build the Argument through Chapters

A number of the books divide their argument into chapters. These books embrace the dramatic change in attitude towards adaptation—from the architects and designers that intervene within the buildings, the developers who have begun to appreciate the value inherent within the already-built, to the legislators who have realised the importance of continuity to the mental and physical wealth of a community.

Within these publications, the chapters are generally stand-alone and can be read as individual discussions, however, the books do tend to construct this as a narrative or journey; so, Stone's *UnDoing Buildings* [12] starts with the strategic approach, advances through peripheral, yet influential, issues such as conservation and installation art, towards resolution within the details. Scott's *On Altering Architecture* [4] commences by constructing the case for reuse, and each chapter reinforces this before the book concludes with some resolutions. Wong's book *Adaptive Reuse: Extending the Lives of Buildings* [35] contains 15 informed chapters that each tackle a different aspect of the subject but, equally and individually, each makes the case for reuse, while Plevoets and Van Cleempoel [7] regard themselves as problem-solvers whose very well-informed survey of the subject provides the motivation for the organisation.

Possibly the most romantic in this collection is *On Altering Architecture*, by Fred Scott, published by Routledge in 2008. Scott had already an established reputation for bringing a radical and intellectual approach to his teaching of interiors as the leader of the programme at Kingston University, and rumours of this book circulated long before publication—so it was eagerly awaited [4] (p. xiv). Scott locates adaptive reuse within a wider cultural framework; he places the subject with art conservation, the search for authenticity, the nature of the copy and the reproduction, the ruins of modernity, and, importantly, he exposed the transgressive nature of remodelling, therefore, moving the subject from beneath the authority of the assured architect towards the more disruptive nature of the designer or artist.

Scott speaks with the authority of long academic experience combined with deep knowledge. He develops a sound theoretical underpinning for the subject, the argument, which is developed over 12 chapters, begins with a call to move away from the scarifying process of conservation towards the progressive, or even transgressive, attitude of adaptation [4] (p. 15). The book continues with discussions of attitudes and practices, and it concludes with resolutions—against pastiche and gratuitous improvement [4] (p. 167). All

buildings are "*in an imperfect state*" and, therefore, meaning, he suggests, can be created through the "*play between the new occupation and the original use*" [4].

*Adaptive Reuse: Extending Lives of Buildings* (Birkhäuser, 2017), by Lilian Wong, is a collection of knowledgeable discussions, it is stylishly produced, and it holds a wide-ranging collection of examples. The book is deliberately provocative, and, although erudite, it is also engagingly angry—angry about the reuse of plunder [35] (It is worth noting that given the colonial connotations connected with the term spolia, in the uncompromising discussion of plunder, Wong is careful to never use that term), about lack of considered care for ancient monuments [35] (p. 90), the exploitative nature of facadism [35] (p. 116), tax incentives as the driver for reuse [35] (p. 55), the dominance of the intervention over the impassive host [35] (p. 174), the false historicism produced by zealous preservation [35] (p. 216), and so on. The architect or designer must negotiate a path, she argues, between Frankenstein-like creations of a self-interested monster [35] (p. 244) and that of the compassionate role of the second violinist—supporting the melody of the host building [35] (p. 246).

At the very beginning of the book is a list of quotations alphabetically classified by their focus. The inclusion of this is both innovative and witty [35] (pp. 13–28), so, Ruskin, Douglas, and ICOMOS are all cited under Repair; Watson, and the British Standards Institution under Addition; and the U.S. Department of the Interior and Eugene Viollet-le-Duc under Restoration. The very early statement that Carlo Scarpa's Castelvecchio is "*timeless*" [35] (p. 6) is questionable. It is possible to contend that although it is, undoubtedly, a work of genius and definitely ground-breaking, adaptive reuse has evolved from Scarpa's contrast and analogy approach to one that pursues the concept of "wholeness" [12] (pp. 183–197).

*UnDoing Buildings* by Sally Stone is a comprehensive study of adaptive reuse that begins with an overview, travels through a discussion about a developed methodology for adaptation, discusses the influence that peripheral areas such smartness, spatial agency, and conservation possess before concluding with an examination about the detailed understanding that the process develops. The book is comprehensive, thorough, and informed. It collects disparate influences and collates them into an organised and influential argument. The publication makes it clear that the process is intrinsically sustainable; that the three tenets of sustainability are a fundamental part of adaptive reuse. It is built upon the urgent need for densification—unlimited horizontal development is no longer ecologically acceptable, therefore, the built environment must learn how to build in on itself, to become more dense, more compact, and more productive.

*Adaptive Reuse of the Built Heritage: Concepts and Cases of an Emerging Discipline* (Routledge, 2019), by Bie Plevoets and Koenraad Van Cleempoel, makes a very well-informed argument for the primacy of the discipline, which has become "*increasingly important as an urban, architectural, and conservation strategy*" [7] (p. 1). The survey of the historical background and strategic approaches is encompassing, but, in reality, the authors are romantics. Theirs is a search for authenticity, for the fundamental poetry inherent within the patina of time and place. Traces, tradition, and empathy activate the "*creative moment of transformation*" [7] (p. 99), thus, they contend, the patina of the palimpsest evolves into an essential part of the design methodology. Plevoets and Van Cleempoel manage to combine both systems; the first half of this extremely influential book is composed of five chapters of built discussion, while the second half is 19 case studies.

> *I believe a lot in the revelatory capacity of reading. . . if one is able to interpret the meaning of what has remained engraved, not only does one come to understand when this mark was made and what the motivation behind it was, but one also becomes conscious of how the various events that have left their mark have become layered, how they relate to one another and how, through time, they have set off other events and have woven together our history.*

> Giancarlo de Carlo, 1990 [36]

## 11. Conclusions

It is obvious that adaptive reuse is no longer regarded as a difficult, undesirable approach but has moved to the centre ground of the development of the built environment. By the third decade of the 21st century, it is recognised as inherently sustainable, as a healthy, friendly, and economically beneficial approach to the development of the built environment. Adaptive reuse addresses the zeitgeist of the 21st century—the imperative to provide for the basic needs of everyone without damaging the planet, to stop uncontrolled horizontal development, to embrace different ideas and cultures, and to understand the importance of environmentally sound development. It has become the expected approach rather than the exception, the first thought rather than the last resort.

The issues of memory and anticipation that drove the contextual movement have had a direct influence on the evolution of a theory of adaptive reuse. These have encouraged architects and designers to embrace a pluralistic agenda that encompasses the anticipated needs and aspirations combined with an understanding of place. The architect reads the qualities of the building and hears what it has to say.

This emphasis upon the distinct qualities of adaptive reuse has coincided with the rise of diversity in architecture—a subject that has evolved far from le Corbusier's aphorism "*masterly, correct, and magnificent play of masses brought together in light*" but now encompasses much greater scope. The diverse foundations of the subject have allowed an expansive attitude that is more inclusive, embraces difference, is sustainable, but is also creative, technologically advanced, and radical. There is now an expectation that the subject is taught in schools of architecture. Adaptive reuse provides a tangible reality in a world that is increasingly distorted by digital interactions and the rise of AI.

The quantity of these writings reflects the position of the subject within the building industry, and beyond that into wider cultural society, and, as the discussions about adaptive reuse have matured, so the scope is moving. It is becoming less Western-centric, technology is developing, and sustainability in all its forms is directly influencing the evolution of the subject. However, as the discussion about the subject has evolved, so these distinct, pluralistic influences have remained. The contextual base for adaptive reuse, combined with an understanding of the needs and aspirations of the users, has proved to be the starting point for discussion and design.

**Funding:** This research has received no external funding.

**Conflicts of Interest:** The authors declare no conflict of interest.

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
