# Peer review of "Notes towards a Definition of Adaptive Reuse"

_2673-8945, doi:10.3390/architecture3030026_

Round 1

Reviewer 1 Report

I have reviewed this essay about the definition of adaptive reuse. The manuscript seems to review articles or a collection of book reviews.

The concept of this essay is interesting because it is an issue not usually discussed objectively and less than that scientifically.

That said, if it is indeed a research article, this article has several shortcomings to be considered a contribution to knowledge. Moreover, it is a collection of opinions and subjective remarks that do not lead anywhere.

At first, a rather vague attempt to define reuse falls short of hermeneutics or other philosophical investigation, which is perhaps revealed or suggested by the quote of Walter Benjamin at the beginning of the article.

There is no perceived methodology in the manuscript.  It seems more like a lengthy introduction to something that does not materialize.

I need help finding in this text definite results, a discussion on some results, or a conclusion.

So, the development of the article is ever-promising but never realized, like a braid or yarn. The introduction is treated with excessive detail, but eventually nothing seems to happen.

The authors may possess some knowledge on this matter, but they must convey it to potential readers other than witty remarks.

There is a total absence of graphs, drawings, or pictures, which seems odd for a Journal on Architecture. There is no explanation for such an apparent flaw.

The authors have not performed a significant effort to bring forth their perhaps valid ideas.

They should identify or trace some methodology and get objectively shaped results.

The discussion also seems lacking and leaves me wondering how we can advance in the everyday and badly needed task of reuse (of buildings).

The quality of English is, on the contrary, very good.

Due to the former findings, the outcome of the manuscript could be more consistent and fairly conclusive, as mentioned above.

Summary of evaluation: This manuscript is promising but needs thorough corrections and further developments to advance to the next phase.

Author Response

Thank you for the review of the submission. I have considered your comments and made a number of revisions.

The suggestion that this is not a research article is very apposite, I agree and have changed the classification to essay.

The vague definition of adaptive reuse has been removed; it was not necessarily relevant to the discussion.

The methodology is a review of literature, this should now be clearer.

The conclusion and the abstract have been strengthened, rendering them more relevant to the discussion.

The witty remarks, which were misguidedly intended to create a light-hearted contribution, have been removed to enable the focus of the discussion to be clearer.

Reviewer 2 Report

The article is well-written, engaging, and presents a helpful exploration of books and articles regarding the theme of Adaptive Reuse.

The text has a series of typos that need to be corrected, inconsistent use of capital letters for the term Adaptive Reuse, and contracted verb forms that should be avoided. Also, some notes do not provide any further explanation and should be revised.

It would be also important to use full names when introducing architects for the first time, as it would be helpful to include names of architects, locations and years of construction when introducing buildings.

Furthermore, the abstract and conclusion should be more specific to the article's contents, referring more clearly to the literature reviewed. The conclusion should provide more insights and critical perspectives considering the books analysed, in particular the most recent ones.

Finally, there is an error in describing the Roma Interrotta project, as this was invented and organised by Pietro Sartogo, and not Michael Graves.

Author Response

Thank you for your review of the submission and your comments about it.

The paper has been thoroughly edited and the typos removed.

The architects, designers, artists and authors are fully named, the years and places added.

The conclusion and the abstract have been strengthened, rendering them more relevant to the discussion.

Thank you for the correction about the Roma Interrotta project. I have read the AD journal several times and had completely missed this fact. It is important.

Round 2

Reviewer 1 Report

The authors have satisfied all my requests.

Accept in present state.